# LRRK2 Knockout Confers Resistance in HEK-293 Cells to Rotenone-Induced Oxidative Stress, Mitochondrial Damage, and Apoptosis

**DOI:** 10.3390/ijms241310474

**Published:** 2023-06-22

**Authors:** Diana Alejandra Quintero-Espinosa, Sabina Sanchez-Hernandez, Carlos Velez-Pardo, Francisco Martin, Marlene Jimenez-Del-Rio

**Affiliations:** 1Neuroscience Research Group, Institute of Medical Research, Faculty of Medicine, University of Antioquia, University Research Headquarters, Calle 62#52-59, Building 1, Laboratory 411/412, Medellin 050010, Colombia; dalejandra.quintero@udea.edu.co (D.A.Q.-E.); calberto.velez@udea.edu.co (C.V.-P.); 2Genomic Medicine Department, Centre for Genomics and Oncological Research (GENYO), Pfizer-University of Granada-Andalusian Regional Government, Parque Tecnólogico Ciencias de la Salud, Av. de la Ilustración 114, 18016 Granada, Spain; sabina.sanchez@genyo.es (S.S.-H.); francisco.martin@genyo.es (F.M.); 3Biochemistry and Molecular Biology 3 and Immunology Department, Faculty of Medicine, University of Granada, Avda. de la Investigacion 11, 18071 Granada, Spain

**Keywords:** apoptosis, clustered regularly interspaced short palindromic repeats, gene edition, human embryonic kidney cell line 293, knockout, Leucine-rich repeat kinase 2, oxidative stress, rotenone

## Abstract

Leucine-rich repeat kinase 2 (LRRK2) has been linked to dopaminergic neuronal vulnerability to oxidative stress (OS), mitochondrial impairment, and increased cell death in idiopathic and familial Parkinson’s disease (PD). However, how exactly this kinase participates in the OS-mitochondria-apoptosis connection is still unknown. We used clustered regularly interspaced short palindromic repeats (CRISPR)/Cas9 LRRK2 knockout (KO) in the human embryonic kidney cell line 293 (HEK-293) to evaluate the cellular response to the mitochondrial inhibitor complex I rotenone (ROT), a well-known OS and cell death inducer. We report successful knockout of the LRRK2 gene in HEK-293 cells using CRISPR editing (ICE, approximately 60%) and flow cytometry (81%) analyses. We found that HEK-293 LRRK2 WT cells exposed to rotenone (ROT, 50 μM) resulted in a significant increase in intracellular reactive oxygen species (ROS, +7400%); oxidized DJ-1-Cys^106^-SO_3_ (+52%); phosphorylation of LRRK2 (+70%) and c-JUN (+171%); enhanced expression of tumor protein (TP53, +2000%), p53 upregulated modulator of apoptosis (PUMA, +1950%), and Parkin (PRKN, +22%); activation of caspase 3 (CASP3, +8000%), DNA fragmentation (+35%) and decreased mitochondrial membrane potential (ΔΨm, −58%) and PTEN induced putative kinase 1 (PINK1, −49%) when compared to untreated cells. The translocation of the cytoplasmic fission protein dynamin-related Protein 1 (DRP1) to mitochondria was also observed by colocalization with translocase of the outer membrane 20 (TOM20). Outstandingly, HEK-293 LRRK2 KO cells treated with ROT showed unaltered OS and apoptosis markers. We conclude that loss of LRRK2 causes HEK-293 to be resistant to ROT-induced OS, mitochondrial damage, and apoptosis in vitro. Our data support the hypothesis that LRRK2 acts as a proapoptotic kinase by regulating mitochondrial proteins (e.g., PRKN, PINK1, DRP1, and PUMA), transcription factors (e.g., c-JUN and TP53), and CASP3 in cells under stress conditions. Taken together, these observations suggest that LRRK2 is an important kinase in the pathogenesis of PD.

## 1. Introduction

The leucine-rich repeat kinase 2 (*LRRK2*) gene (Gene ID: 120892) encodes a multifunctional protein composed of a GTPase domain, four scaffold domains (Armadillo, Ankyrin, LRR, WD) and a kinase domain [1,2]. This kinase has been associated with the late-onset autosomal dominant type of Parkinson’s disease [3,4,5,6]. Although the specific molecular mechanism leading to dopaminergic (DAergic) neuronal death in Parkinson’s disease (PD) is not yet fully understood [7,8], studies have demonstrated that LRRK2 is associated with increased susceptibility to oxidative stress (OS), mitochondrial depolarization, and cell death [9,10,11,12,13,14]. While the LRRK2 kinase domain has been recognized to be responsible for OS-induced neurotoxicity [10,14,15], it is not yet clear how the lack of LRRK2 protein might affect mitochondria-associated proteins such as mitochondrial import receptor translocase of the outer membrane of mitochondria (TOM20), PTEN-induced kinase 1 (PINK1), and Parkin (PRKN) when cells are challenged with specific agents that alter normal mitochondrial physiology. Therefore, the study of LRRK2 and its neurotoxin interactions might be vital for understanding the biology of neuronal deterioration in PD [16].

Previous work by our laboratory has shown that rotenone (ROT), a class 5 mitocan-specific inhibitor of mitochondrial complex I [17,18], induces apoptosis in nerve-like cells (NLCs) through an OS mechanism concomitant with LRRK2 phosphorylation at serine ^935^ an [15]. Interestingly, it has been shown that LRRK2 is activated by H_2_O_2_ in the human embryonic kidney cell line 293 (HEK-293) and in primary cortical neuron cultures from mice [19,20]. Since LRRK2 plays a role in PD [20], we hypothesized that by rendering the LRRK2 gene inoperative, the cells might become resistant to ROT-induced OS, mitochondrial damage, and apoptosis in vitro.

To test this assumption, HEK-293 LRRK2 wild-type (WT) and knockout (KO) cells obtained by the clustered regularly interspaced short palindromic repeats (CRISPR)/associated protein Cas9 system [21] were left untreated or treated with ROT (50 μM). We found that LRRK2 WT cells, but not KO cells, exposed to ROT resulted in a significant increase in intracellular reactive oxygen species (ROS), oxidized DJ-1-Cys^106^-SO_3_-, p-Ser^935^-LRRK2, p-Ser^65^-c-JUN, high expression of tumor protein p53 (TP53), TP53 upregulated modulator of apoptosis (PUMA), PINK1, PRKN, activation of caspase 3 (CASP3), DNA fragmentation, decreased mitochondrial membrane potential (ΔΨm), and colocalization of fission protein GTPase dynamin-related protein 1 (DRP1) with TOM20. Taken together, these observations suggest that LRRK2 might be implicated in the pathogenesis of PD by acting as a proapoptotic kinase by regulating mitochondrial proteins (e.g., PRKN, PINK1, DRP1, and PUMA), transcription factors (e.g., c-JUN and TP53), and CASP3.

## 2. Results

### 2.1. LRRK2 Gene Edition by Clustered Regularly Interspaced Short Palindromic Repeats (CRISPR)/Cas9 Method in Human Embryonic Kidney Cell Line 293 (HEK-293) Cells

Previous studies have shown that HEK-293 cells are amenable to genetic editing by the CRISPR/Cas9 method [22], specifically on the LRRK2 gene [20]. Therefore, we used CRISPR-Cas9 technology to obtain LRRK2 knockout (hereafter LRRK2 KO) in HEK-293 cells. To achieve this aim, three sgRNAs targeting the start codon (ATG) exon 1 of LRRK2 (e.g., 1. Fw sgRNA (gL1), 2. Rv sgRNA (gL2), and 3. Rv sgRNA (gL3), Appendix A, green color) with their respective protospacer adjacent motif (PAM, red color) and Cas9 cleavage site (blue arrowheads) were designed using CRISPOR software 3.1 (Appendix A). After nucleofection with RNP/sgRNA LRRK2 complexes, PCR amplicons from WT and edited LRRK2 fragments (Appendix A) were obtained by LRRK2-specific primers (Appendix A). The sequences of the PCR products were analyzed by the Inference of CRISPR Editing (ICE) tool provided by Synthego^®^. Indeed, HEK-293 LRRK2 KO cell editing was confirmed by Sanger sequencing (Appendix A–C), the discordance plot (Appendix A–F), and indel plot analysis (Appendix A–I, Appendix A). LRRK2 gene disruption by CRISPR/Cas9 decreased total LRRK2 expression by −81% in LRRK2 KO (gL3, Appendix A–L) compared to WT HEK-293 LRRK2 cells according to flow cytometry analysis (Figure 1A–C). Similar results were obtained by fluorescent microscopy (FM, Figure 1D–J).

### 2.2. HEK-293 LRRK2 Knockout (KO) Cells Are Resilient to Cell Stress

We assessed whether ROT induced apoptosis in both WT HEK-293 cells and KO cells. HEK-293 LRRK2 WT and KO cells were exposed to increasing concentrations of ROT (1, 5, 10, 50 μM) for 6 h. As shown in Figure 2, ROT induced a high percentage of depolarized mitochondria (Figure 2A,C) and generated a high percentage of ROS (Figure 2D,F) and fragmented DNA (Figure 2G,I) in HEK-293 LRRK2 WT cells. In contrast, ROT caused a moderate loss of ΔΨm (Figure 2B,C, −58% reduction) but neither a significant generation of ROS (Figure 2E,F) nor nuclear fragmentation (Figure 2H,I) in LRRK2 KO cells. Similar data were obtained by FM analysis (Figure 2J–AA). Since ROT (50 μM) was the most effective concentration for inducing mitochondrial depolarization, oxidative stress (OS), and nuclear fragmentation in LRRK2 WT cells, but did not affect LRRK2 KO cells, we selected this concentration for future experiments.

The above observations impelled us to determine the redox status of the stress sensor protein DJ-1-Cys^106^-SH [23], which is specifically oxidized by H_2_O_2_ [24]. As shown in Figure 3, ROT induced an important increase (+53%) in Cys^106^-SO_3_ (-sulfinyl) in HEK-293 LRRK2 WT cells (Figure 3A,C). Interestingly, ROT induced an increase (+33%) in Cys^106^-SO_3_ in KO cells (Figure 3B,C). Similar data were found by FM examination (Figure 3D–P). 

Then, we determined whether ROT induced LRRK2 phosphorylation [15] in both LRRK2 WT and KO cells. Cytometry analysis revealed that ROT increased p-S^935^-LRRK2 by +70% in HEK-293 LRRK2 WT cells compared to untreated cells (Figure 4A,C), whereas ROT induced no significant LRRK2 phosphorylation in LRRK2 KO cells (Figure 4B,C). Similar information was found by FM (Figure 4D–P).

### 2.3. LRRK2 KO Inhibits Rotenone (ROT)-Induced Upregulation of PTEN-Induced Kinase 1 (PINK1)/Parkin (PRKN)

We further determined the effect of this neurotoxin on the mitochondrial import receptor translocase of the outer membrane of mitochondria (TOM20) translocase, PTEN- induced kinase 1 (PINK1), and parkin (PRKN) in HEK-293 cells. Figure 5 shows that ROT caused no effect on TOM20 in LRRK2 WT (Figure 5A,C) and LRRK2 KO cells (Figure 5B,C). However, ROT induced a differential effect on PINK1 and PRKN. While ROT provoked a significant increase in PRKN by 22% in HEK-293 LRRK2 WT cells (Figure 5D,F) and PINK1 by 49% (Figure 5G,I) and 83% (Figure 5H,I) in LRRK2 WT and KO cells, respectively, the expression of PRKN diminished by −23% in HEK-293 LRRK2 KO cells (Figure 5E,F). Similar data were obtained by FM evaluation (Figure 5J–AI).

### 2.4. LRRK2 KO Blocks Rotenone (ROT)-Induced Dynamin-Related Protein 1 (DRP1)/Translocase of the Outer Membrane of Mitochondria (TOM20) Colocalization

Several lines of evidence suggest that dynamin-related protein 1 (DRP1) is a mediator of mitochondrial fission and apoptosis [25,26] through phosphorylation of LRRK2 [27]. Therefore, we assessed whether cytoplasmic DRP1 translocates to mitochondria in HEK-293 cells under ROT treatment. WT and KO cells were exposed to ROT, and DRP1 and TOM20 were evaluated. As shown in Figure 6, DRP1 colocalized with TOM20 in the mitochondria of HEK-293 LRRK2 WT cells exposed to ROT (Figure 6A–F vs. Figure 6G–L), whereas it localized in the cytoplasm in LRRK2 KO cells (Figure 6M–R vs. Figure 6S–X). These observations were consistent with the percentage analysis of the colocalized area (Figure 6Y).

### 2.5. LRRK2 KO Inhibits Rotenone (ROT)-Induced Apoptosis

To determine whether ROT induced apoptosis in both WT and KO cells, cells were exposed to ROT. Then, we evaluated the activation of proapoptotic proteins such as transcription factors (e.g., c-JUN, TP53), PUMA, and executer protein caspase 3 by flow cytometry [28]. As shown in Figure 7, ROT increased the activation of p-Ser65-cJUN (Figure 7A,C), TP53 (Figure 7D,F), PUMA (Figure 7G,I), and CASP3 (Figure 7J,L) by +171%, +2000%, +1950%, and +8000%, respectively, in HEK-293 LRRK2 WT cells compared to untreated WT cells. However, ROT did not induce appreciable expression of apoptotic markers in LRRK2 KO cells compared to untreated KO cells (Figure 7B,C,E,F,H,I,K,L). Similar observations were revealed by FM (Figure 7M–BL).

## 3. Discussion

We report for the first time that HEK-293 cells devoid of the LRRK2 kinase protein were resistant to ROT-induced OS, mitochondrial damage, and apoptosis. LRKK2 KO cells exposed to ROT show almost no signs of intracellular generation of ROS, according to the DCF+ cell analysis, or oxidation of the stress sensor protein DJ-1-Cys^106^-SO_3_. Additionally, LRRK2 KO cells show neither an important loss of ΔΨm, significant activation of the transcription factors c-JUN and TP53, activation of the pro-apoptotic BH3-only protein PUMA, the executer CASP3, nor nuclear fragmentation. Outstandingly, all markers of OS and apoptosis were concomitantly absent with unphosphorylated p-S^935^-LRRK2. Furthermore, we found that the maintenance of mitochondria and the antioxidant protein PRKN were significantly downregulated in LRKK2 KO cells, while the PRKN activator protein PINK1 was upregulated. Moreover, DRP-1 was primarily localized in the cytoplasm, and the mitochondrial protein TOM20 was unaffected.

To support the above observations, when HEK-293 LRRK2 WT cells were untreated or treated with ROT, the opposite phenotypic responses to KO cells were observed. Indeed, HEK-293 LRRK2 WT cells exposed to ROT had a significant increase in intracellular ROS via inhibition of mitochondrial complex I [29,30,31], as revealed by a high percentage of DCF+ cells and oxidized DJ-1Cys^106^-SO_3_, which is specifically oxidized by H_2_O_2_ [24]. In agreement with others [19,20], we found that ROT/H_2_O_2_ activated LRRK2 kinase, as evidenced by a high percentage of p-S^935^-LRRK2. However, how H_2_O_2_ activates LRRK2 kinase is still unknown. One possibility is that H_2_O_2_ activates IKK [32] through activation of the MEKK1/IKK complex [33]. Once IKK is active, it can in turn activate LRRK2 kinase by phosphorylation at residues Ser^910^ and Ser^935^ [34]. This last phosphorylated residue (Ser^935^) was used in our present work as a positive surrogate marker for LRRK2 activity. Whatever the mechanism, we demonstrated that ROT/H_2_O_2_ activates LRRK2 kinase. After activation, LRRK2 might contribute to apoptosis signaling by either activating apoptosis signaling kinase-1 (ASK-1, [35]) or triggering the pro-apoptogenic protein c-JUN through MKK4/MAPK kinase-JNK [36,37].

Mitochondria are critical organelles in the decision-making process regarding cell life and death [38]. Importantly, PRKN, an E3 ubiquitin ligase [39], and PINK1 have been demonstrated to mediate mitochondrial quality control [40,41]. Several studies have shown that upon a loss of ΔΨm, cytosolic PRKN is recruited to mitochondria by PINK1 [42,43], which is followed by stimulation of mitochondrial autophagy [44]. Moreover, it has been shown that LRRK2 attenuates PINK1/PRKN-dependent mitophagy in a kinase-dependent manner, contributing to apoptosis [45], a phenomenon that might be deficient in LRRK2 KO cells, thereby increasing the survival and resistance of cells against OS stimuli. In line with these observations, we found that HEK-293 LRRK2 WT cells exposed to ROT displayed high expression of both PRKN and PINK1, whereas in the absence of LRRK2 and H_2_O_2_, PRKN was concomitantly downregulated with an unaltered mitochondrial potential, probably due to its high mitochondrial PRX3 activity [11].

Although we were unable to detect any increase or decrease in the expression of mitochondria-associated proteins such as TOM20 in LRRK2 WT or LRRK2 KO cells, others have provided evidence that LRRK2 interferes with protein-protein interactions involving PRKN on the outer mitochondrial membrane (OMM) and that these interactions can be rescued by the inhibition of LRRK2 kinase activity [45]. Moreover, Neethling and coworkers [46] reported that LRRK2 is associated with and colocalized with subunits of the TOM complex, either under basal or stress-induced conditions. However, whether these LRRK2-induced interactions are triggered by ROT in HEK-293 cells requires further investigation. Whatever the mechanism of PINK1/PRKN/TOM, we found that ROT-induced mitochondrial depolarization in LRRK2 WT concomitantly appeared with a significant decrease in the expression of PRKN, a phenomenon not detected in LRRK2 KO cells under OS.

Interestingly, both the transcription factors c-JUN [47] and TP53 [48,49] regulate PUMA, a proapoptotic protein directly implicated in mitochondrial depolarization and cell death [50]. To further aggravate matters, LRRK2 also phosphorylates TP53 [49,51], thereby stabilizing the protein and amplifying the apoptosis signal. We also found that the cytoplasmic fission protein DRP1 colocalized with the TOM20 protein. This observation suggests that DRP1 can translocate from the cytoplasm to mitochondria, probably due to LRRK2 phosphorylation at residue Thr^595^ [27], further contributing to cell death. Taken together, these observations imply that active (p-Ser^935^) LRRK2 kinase is strongly connected with the phosphorylation of key proteins involved in the activation of apoptosis (e.g., ASK-1, MEKK1, TP53, and DRP1) in HEK-293 cells under stress stimuli (e.g., ROT or H_2_O_2_). Noticeably, the absence of OS and apoptosis markers in HEK-293 LRRK2 KO cells subjected to stress further strengthens our premise that LRRK2 acts as a proapoptotic driver of protein kinases.

In agreement with others, our data suggest that ROT induces apoptosis in HEK-293 LRRK2 WT cells through activation of CASP3 [52] and subsequent CASP3-dependent nuclei fragmentation [53,54]. Clearly, these phenotypes were not observed in HEK-293 LRRK2 KO cells or in dopaminergic (DAergic) neurons derived from marmoset embryonic and iPSCs where LRRK2 was truncated [55]. Taken together, these observations suggest that LRRK2 might be implicated in the ROS-induced activation of CASP3 and apoptosis. Furthermore, our findings imply that LRRK2 WT under OS stimuli might biochemically behave like the LRRK2 Gly2019Ser mutant in DAergic neurons, wherein the LRRK2 mutant mediates increased intracellular ROS and decreased neuronal viability [10].

Recently, Deshpande and co-workers [56] have shown that pharmacological inhibition of LRRK2, Lrrk2 knockdown (KD), or Lrrk2 KO all lead to increased translation. Moreover, they found that treatment of midbrain cultures with ROT (1 nM) increased LRRK2 activity, as assessed by increased phosphorylation of S^935^-LRRK2, ROT reduced translation by 40% in dopaminergic neurons, and this was prevented by the LRRK2 inhibitor MLi-2. These results indicate that protein synthesis is repressed in a LRRK2-dependent fashion in cellular models of PD. In agreement with those findings, we found that ROT increased phosphorylation of S^935>^-LRRK2 in HEK-293 LRRK2 WT cells. However, whether RNA translation is repressed in HEK-293 LRRK2 WT cells treated with ROT or LRRK2 KO cells requires further investigation. We anticipated that HEK-293 LRRK2 WT cells exposed to ROT would show a significant decrease in RNA translation, but this effect would be increased in LRRK2 KO cells. Interestingly, Ünal and colleagues [57] have reported that ROT (5 μg/L, approximately 13 nM) for 4 weeks induced a significant increase in the expression of *lrrk2* and *dj-1*, but a significant decrease in the expression of *prkn*, *pink1*, and *bdnf* (brain-derived neurotrophic factor), according to reverse transcription (cDNA synthesis) and quantitative real-time PCR in zebrafish (*Danio rerio*). However, further experiments are needed in either mammalian dopaminergic (DAergic) or non-DAergic cells (e.g., HEK-293 cells) to validate those findings.

Rotenone produces selective dopaminergic neurotoxicity through complex I inhibition, followed by OS generation and resultant oxidative damage [58]. Therefore, ROT has commonly been used as a model neurotoxin in vitro and in vivo in different experimental settings to model PD [59]. However, the neurotoxic selectivity is dose- and time-dependent, where concentrations < 40 nM have been shown to promote specific neurotoxicity in primary cortical neuronal culture, e.g., 20–30 nM ROT for 11–24 h induced selective dopaminergic neurotoxicity without affecting other mesencephalic cells [52,60,61], or ROT concentrations as low as 1 nM can induce selective dopaminergic toxicity in neuron and glia primary rat culture after 8 days of exposure [62]. These findings coincide with the in vivo data and show that rotenone selectively targets dopaminergic neurons [59]. Unlike primary cultures, immortalized neuronal cell lines are more resistant to rotenone toxicity. Therefore, the doses of ROT can increase several orders of magnitude, e.g., >330–1600 folds. For instance, the human dopaminergic SH-SH5Y neuroblastoma cells [63], the Lund human mesencephalic (LUHMES) cell line [64], iPSC-derived human dopamine neurons [65], as well as rat adrenal pheochromocytoma PC12 cells [66], and rat N27 dopaminergic cells [67] have been exposed to 10–50 μM ROT for different periods of time. In our experimental approach, we exposed the non-dopaminergic HEK-293 LRRK2 WT and KO cells to 50 μM for 6 h, which is a comparable condition to those reported in previously mentioned dopaminergic cell lines. Furthermore, the healthy donor origin of HEK-293 cells makes these cells an attractive tool for studying the effects of environmental exposures; e.g., ROT leading to mitochondrial dysfunctions and/or a PD-like phenotype. Therefore, HEK-293 cells are a suitable and valid model system to study the cellular and molecular aspects of PD.

In the context of the present investigation, HEK-293 cells offer several advantages as an in vitro model system to study PD. Technically, they are easy to handle, rapidly grow, can be readily transfected, and are amenable to stringent quantitative assessments. Developmentally, HEK-293 cells and neurons originate from the same precursor line; this means that the fundamental biological processes and their regulatory mechanisms (i.e., transcription, translation, protein folding and trafficking, and so on) are quite similar but not identical. Indeed, the HEK-293 cells generated in the 1970s from normal primary human embryonic kidney (HEK) cells with sheared adenovirus 5 DNA show typical features of immature neurons, such as the expression of the neurofilament (NF) subunits NF-L, NF-M, NF-H, alpha-internexin, vimentin, keratins 8 and 18, and also revealed expression of mRNAs specific for numerous other genes normally expressed in neuronal lineage cells [68]. Based on these observations, HEK-293 can be considered a human neuronal cell line model. Thus, HEK293 cells provide a reasonable approximation for addressing numerous questions of basic biology in PD. Specifically, since LRRK2 is a well-conserved evolutionary gene [69], HEK-293 cells have been used to identify molecular substrates of this kinase [70] and to study LRRK2 mutations’ functional analysis (e.g., refs. [71,72]). Furthermore, given that HEK-293 cells have demonstrated a clear pro-apoptotic transcriptional response profile in neurons undergoing apoptosis [73] and displayed elements of autophagy mechanistically similar to those expressed in DAergic neurons [74], these observations suggest that HEK-293 cells might be suitable for cellular and molecular studies in PD as reported in the present work. However, HEK-293 cells present at least two main limitations. First, HEK-293 cells have been resilient to their transdifferentiation into dopaminergic neurons. Second, these cells do not endogenously express dopamine lineage markers such as tyrosine hydroxylase (TH), which catalyzes the hydroxylation of L-tyrosine into L-DOPA (l-3,4-dihydroxyphenylalanine), the rate-limiting step in the synthesis of dopamine (DA), or the human dopamine transporter (hDAT). Of note, however, is that this limitation in a specific scientific context might turn out to be an advantage. Indeed, HEK-293 cells have been used to ectopically express not only hDAT to study, e.g., the toxic effect of MPTP [75], but also dopaminergic receptors, e.g., D1 [76] and/or D5 [77]. Finally, the choice of a specific cellular model such as HEK-293 cells entails the requirement to focus on one aspect of the disease, e.g., the effect of ROT on LRRK2 WT and KO, while ignoring others (e.g., the effects of astroglia and microglia on neurons exposed to ROT as it occurs in vivo [78]).

## 4. Materials and Methods

### 4.1. HEK-293 Cell Line

HEK-293, a specific immortalized cell line derived from a human embryonic kidney, was purchased from AcceGen Biotech (cat# ABC-TC0008, AcceGen Biotech, Fairfield, NJ, USA) and cultured according to the suppliers’ recommendations. Briefly, cells were grown in Dulbecco’s Modified Eagle’s Medium (DMEM, cat#D0819, Sigma, St. Louis, MO, USA), supplemented with Fetal Bovine Serum (FBS, cat#CVFSVF00-01, Eurobio Scientific, Paris, France) to a final concentration of 10% in a humidified incubator at 37 °C, supplemented with 5% CO_2_. Growth media was replaced every 2–3 days.

### 4.2. CRISPR/Cas9 Genome Editing of HEK-293 Cells to Produce LRRK2 Knockout Cell Lines

To generate an LRRK2 knockout cell line, a CRISPR/Cas9 genome editing approach was used. A guide RNA (gRNA) targeting exon 1 of the LRRK2 gene (5′-CTGACAGCTGCCACTAGCCA-3′) was designed using the CRISPOR Design 3.1. Tool (http://crispor.tefor.net/ accessed on 15 December 2022). HEK-293 cells were transfected with RNP (Cas9 and gRNA assembled in a 3:1 molar ratio, respectively, provided by Integrated DNA Technologies, Coralville, IA, USA), using a 4D-Nucleofector^®^ (Cat# AAF-1003B, #AAF-1003X, Lonza Inc., Gampel, Gampel-Bratsch, Switzerland). Transfected cells were collected and expanded for polymerase chain reaction (PCR) and DNA sequencing analyses. To confirm gene editing of the LRRK2 gene, a region of exon 1 was PCR-amplified using a forward primer (5′-ATAAACAGGCGGGCGTGGG-3′ Integrated DNA Technologies, Coralville, IA, USA) and a reverse primer (5′-TGCGGCTCCTTAAGAGTCCGG-3′, Integrated DNA Technologies, Coralville, IA, USA), and the resulting PCR product was sequenced. Efficacy of edition and KO score were obtained by Synthego Performance Analysis (ICE Analysis, 2019. v3.0. Synthego, Redwood City, CA, USA) of sanger sequences from edited cells versus wild-type cells, according to ref. [79].

### 4.3. Experiments with HEK-293 WT LRRK2 and LRRK2 Knockout (KO) Cells

#### 4.3.1. Flow Cytometry (FC) Analyses

##### Assessment of Mitochondrial Membrane Potential (∆Ψm)

The analysis of ∆Ψm was performed according to ref. [80]. Briefly, cells (2.5 × 10^5^) were left untreated or treated with increasing concentrations of rotenone (ROT, 1, 5, 10, 25, and 50 μM) for 6 h at 37 °C. Cells were then incubated with the MitoTracker Deep Red compound (5 nM, final concentration, Thermo Fischer Scientific, Waltham, MA, USA) for 20 min at room temperature in the dark. Cells were detached from plates using 0.25% trypsin. Cells were analyzed using a LSRFortessaTM cell analyzer (model # 649225B4, BD Biosciences, San Jose, CA, USA). The experiment was performed 3 times, and 10,000 events were acquired for analysis. Quantitative data and figures were obtained using FlowJo Data Analysis Software (v10 BD Biosciences, San Jose, CA, USA).

##### Determination of DNA Fragmentation

DNA fragmentation was determined as described elsewhere [80]. Cells in the sub-G_0_/G_1_ phase were used as a marker of apoptosis. After treatment, cells (2.5 × 10^5^) were washed twice with PBS (pH 7.2) and stored in 95% ethanol overnight at −20 °C. Then, cells were washed and incubated in a 400 μL solution containing propidium iodide (PI; 50 μg/mL; cat#P4170, Sigma, St. Louis, MO, USA), RNase A (100 μg/mL; cat #1010914200, Sigma, St. Louis, MO, USA), EDTA (50 mM; cat # EX0550-5, Sigma, St. Louis, MO, USA), and triton X-100 (0.2%; cat#93419, Sigma, St. Louis, MO, USA) for 60 min at 37 °C. The cell suspension was analyzed for propidium iodine (PI) fluorescence by using a LSRFortessaTM cell analyzer (model # 649225B4, BD Biosciences, San Jose, CA, USA). DNA fragmentation was assessed three times in independent experiments. Quantitative data and figures from the sub-G_0_ and G_1_ populations were obtained using FlowJo Data Analysis Software (v10 TIBCO^®^Data Science, Palo Alto, CA, USA).

##### Evaluation of Reactive Oxygen Species (ROS)

The ROS (e.g., H_2_O_2_) were determined with 2′,7′-dichlorofluorescein diacetate (1 μM, DCFH2DA Sigma, St. Louis, MO, USA) according to ref. [80]. After cell treatment with compounds of interest, cells (1 × 10^5^) were incubated with a DCFH2-DA reagent for 30 min at 37 °C in the dark. Cells were then washed, and DCF fluorescence was determined using the LSRFortessaTM cell analyzer (model # 649225B4, BD Biosciences, San Jose, CA, USA). The assessment was repeated three times in independent experiments. Quantitative data and figures were obtained using FlowJo 7.6.2 Data Analysis Software.

##### Evaluation of Oxidative Stress and Apoptosis

HEK-293 WT LRRK2 and LRRK2 KO cells were left untreated or treated with ROT (50 μM) for 6 h at 37 °C. Cells were detached from plates using 0.25% trypsin, washed twice with PBS (pH 7.2), and stored in 4% paraformaldehyde overnight at 4 °C. After simultaneous permeabilization and blockage with 0.1% Triton X-100 and 3% BSA, cells were immunostained according to ref. [80]. Briefly, cells were incubated overnight at 4 °C with primary antibodies, anti-p-c-Jun (KM-1) Alexa Fluor^®^594 (cat # sc-822 AF594, Santa Cruz Biotech, Santa Cruz, CA, USA), anti-PUMAα (B-6) Alexa Fluor^®^488 (cat # sc-377015 AF488, Santa Cruz Biotech, Santa Cruz, CA, USA), anti-p53 (DO-1) Alexa Fluor^®^594 (cat # sc-126 AF594, Santa Cruz Biotech, Santa Cruz, CA, USA), and Active Caspase Detection label FITC-VAD-FMK (cat# QIA90, Sigma, St. Louis, MO, USA). Cells were also incubated with anti-DJ1-ox-Cys^106^ (cat #MABN1773, Sigma, St. Louis, MO, USA), anti-LRRK2 (cat #43733, Santa Cruz Biotech, Santa Cruz, CA, USA), anti-p-Ser935 LRRK2 (cat #AB133450, Abcam, Trumpington, Cambridge, UK), anti-DRP1 (cat #ABT155, Sigma, St. Louis, MO, USA), anti-PINK1 (cat #33796, Santa Cruz Biotech, Santa Cruz, CA, USA), and anti-Parkin (cat #30130, Santa Cruz Biotech, Santa Cruz, CA, USA). All primary antibodies were prepared at a final concentration of 2 µg/mL (1:500) in blocking solution (PBS 1×, 1.5% BSA). After several washes, cells with unconjugated antibodies were incubated with either Alexa Fluor 488 donkey anti-goat (cat# A11055, Thermo Fischer Scientific, Waltham, MA, USA) or Alexa Fluor 594 donkey anti-rabbit (cat# A21207, Thermo Fischer Scientific, Waltham, MA, USA) secondary antibodies. The assessment was repeated three times in independent experiments. Quantitative data and figures were obtained using FlowJo v7.6.2 data analysis software. The specificity of primary antibodies is reflected in the specificity of the secondary antibodies provided by Thermo Fisher Scientific (cat #A11055 and cat # A21207, Thermo Fischer Scientific, Waltham, MA, USA). Indeed, the anti-rabbit and anti-goat secondary antibodies are affinity-purified antibodies with well-characterized specificity for rabbit or goat immunoglobulins and are useful in the detection of their specified targets.

#### 4.3.2. Fluorescence Microscopy (FM) Analyses

##### Immunofluorescence Analyses

HEK-293 LRRK2 WT and LRRK2 KO were left untreated or treated as described above. Cells were then fixed with 4% paraformaldehyde for 20 min. After simultaneous permeabilization and blockage with 0.1% Triton X-100 and 5% BSA, cells were incubated overnight at 4 °C with conjugated primary antibodies, anti-p-c-Jun (KM-1) Alexa Fluor^®^594 (cat # sc-822 AF594, Santa Cruz Biotech, Santa Cruz, CA, USA), anti-PUMAα (B-6) Alexa Fluor^®^488 (cat # sc-377015 AF488, Santa Cruz Biotech, Santa Cruz, CA, USA), anti-p53 (DO-1) Alexa Fluor^®^594 (cat # sc-126 AF594, Santa Cruz Biotech, Santa Cruz, CA, USA), and Active Caspase Detection label FITC-VAD-FMK (cat# QIA90, Sigma, St. Louis, MO, USA). Cells were also incubated with anti-DJ1-ox-Cys^106^ (cat #MABN1773, Sigma, St. Louis, MO, USA), anti-LRRK2 (cat #43733, Santa Cruz Biotech, Santa Cruz, CA, USA), anti-p-Ser^935^ LRRK2 (cat #AB133450, Abcam, Trumpington, Cambridge, UK), anti-DRP1 (cat #ABT155, Sigma, St. Louis, MO, USA), anti-PINK1 (cat #33796, Santa Cruz Biotech, Santa Cruz, CA, USA), and anti-Parkin (cat #30130, Santa Cruz Biotech, Santa Cruz, CA, USA). All primary antibodies were prepared at a final concentration of 100 μg/mL (1:200) in blocking solution (1.5% BSA). After several washes, cells incubated with unconjugated antibodies were incubated with either Alexa Fluor 488 donkey anti-goat (Life Technologies, cat # A11055), or Alexa Fluor 594 donkey anti-rabbit (cat # A21207) secondary antibodies. The nuclei staining was carried out using Hoechst 33342 (2.5 μM). The IF analysis was assessed three times in independent experiments.

##### Immunofluorescence Colocalization Analysis of the DRP1 and Translocase of the Outer Mitochondrial Membrane (TOM)20

The immunofluorescent staining procedure was according to the standard procedures described in ref. [81]. Briefly, untreated and treated cells with ROT (50 μM) were plated on a positively charged slide and air-dried. Cells were fixed with 4% formaldehyde for 20 min. After permeabilization, cells were incubated overnight at 4 °C with a primary anti-DRP1 antibody (cat #ABT155, Sigma, St. Louis, MO, USA) in combination with a TOM20 mouse monoclonal antibody (cat #ab115746, Abcam, Trumpington, Cambridge, UK). All primary antibodies were prepared at 1:200. After several washes, cells were incubated with Alexa Fluor488 (cat #A11055, Thermo Fisher Scientific, Waltham, MA, USA), Alexa Fluor 594 donkey anti-rabbit (cat #A21207, Thermo Fisher Scientific, Waltham, MA, USA), or Alexa Fluor 594 donkey anti-mouse IgG (cat #R37115, Thermo Fisher Scientific, Waltham, MA, USA) secondary antibodies according to the supplier’s protocol. The percentage of positive colocalization was calculated as the area that each marker colocalized over the total area of HEK293 LRRK2 WT and LRRK2 KO cells by ImageJ software (v2.9.0 National Health Institute, Bethesda, MD, USA). The merged figures were transformed into RGB color images, and the background was subtracted. The cellular measurement of the colocalized area was obtained by selecting the yellow area (corresponding to the peak of overlapped color channels) and applying the same threshold color for controls and treatments. Additionally, intensity profiles were obtained using ZEN lite v3.4 software (Carl Zeiss Canada Ltd., Toronto, ON, Canada). Experiments were performed in triplicate on three independent experiments.

### 4.4. Photomicrography and Image Analysis

The fluorescent microscopy photographs were taken using a Floyd cells imaging station microscope (cat #4471136, Thermo Fisher Scientific, Waltham, MA, USA). Ten images per well acquired by the Floyd cells imaging station were analyzed by ImageJ software (v2.9.0, National Health Institute, USA). The figures were transformed into 8-bit images, and the background was subtracted. The cellular measurement regions of interest (ROI) were drawn around nuclear (for the case of transcription factors and apoptosis effectors) or overall cells (for cytoplasmic probes), and the fluorescence intensity was subsequently determined by applying the same threshold for controls and treatments. Mean fluorescence intensity (MFI) was obtained by normalizing total fluorescence to the number of nuclei.

### 4.5. Data Analysis

In this experimental design, a vial of HEK293 cells (LRRK2 WT or LRRK2 KO) was thawed, cultured, and the cell suspension was pipetted at a standardized cellular density of 2.3 × 10^5^ cells/cm^2^ into different wells of a 6-well plate. Cells (i.e., the biological and observational units [82]) were randomized to wells by simple randomization (sampling without replacement method), and then wells (i.e., the experimental units) were randomized to treatments by a similar method. Experiments were conducted in triplicate. The data from individual replicate wells were averaged to yield a value of n = 1 for that experiment, and this was repeated on three occasions blind to the experimenter for flow cytometer analysis for a final value of n = 3 [82]. Based on the assumptions that the experimental unit (i.e., the well) data comply with the independence of observations, the dependent variable is normally distributed in each treatment group (Shapiro-Wilk test), and variances are homogeneous (Levene’s test); the statistical significance was determined by Student’s *t*-test, one-way, or two-way ANOVA followed by Bonferroni’s, Tukey’s, or Dunnett’s T3 (if there is not homogeneity of variances) post hoc comparison calculated with GraphPad Prism v5.0 (v5.0, San Diego, CA, USA) or IBM^®^ SPSS^®^ statistics software (v25 Armonk, NY, USA) (Appendix A). Differences between groups were only deemed significant with a *p*-value of <0.05 (*), <0.01 (**), and <0.001 (***). All data are illustrated as the mean ± S.D.

## 5. Conclusions

Our findings suggest that LRRK2 acts as a critical trigger in OS-induced cell death by regulating mitochondrial proteins (e.g., PINK1, PRKN, DRP-1, and PUMA) and transcription factors (e.g., c-JUN and TP53) [83]. Therefore, LRRK2 gene editing [84] or pharmacological treatment with either synthetic [15,85] or natural [86,87] may protect neurons from deterioration in PD. Even though the HEK-293 model helped to clarify the potential role of LRRK2 in the OS-induced apoptosis pathway, further siRNA or shRNA studies in primary neurons are needed to confirm the present findings.

## Figures and Tables

**Figure 1 ijms-24-10474-f001:**
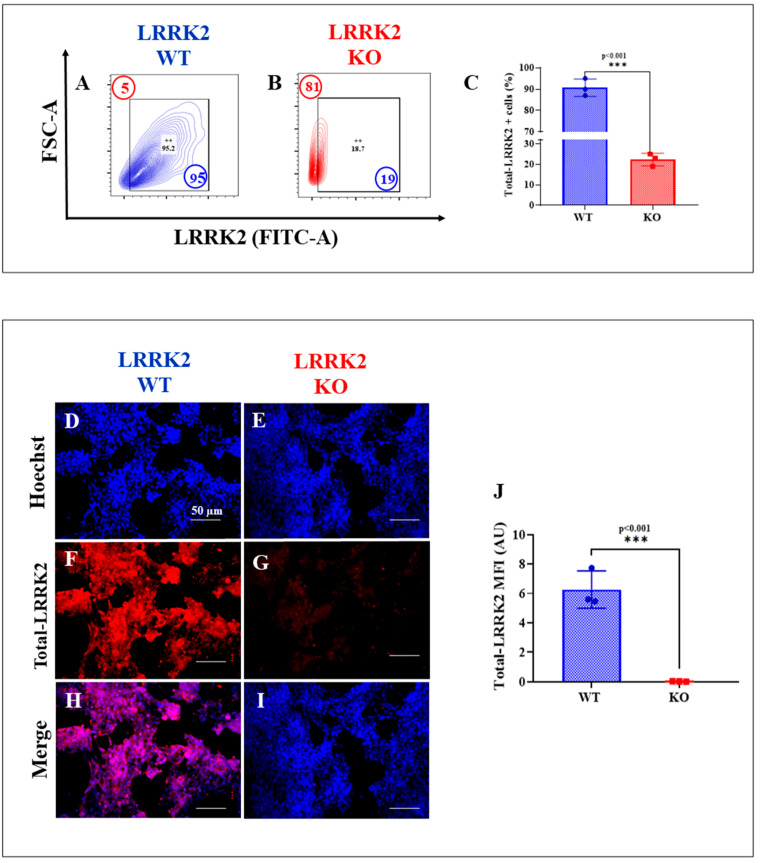
LRRK2 KO decreases the expression of LRRK2 protein. (**A**,**B**) Representative flow cy-tometry contour plots show total LRRK2 protein in LRRK2 WT and LRRK2 KO cells. (**C**) Percent-age of total LRRK2 protein in LRRK2 WT and LRRK2 KO cells. The data are presented as the mean ± SD of three independent experiments (n = 3). Student’s *t*-test: Statistically significant differences when *** *p* < 0.001. Additionally, HEK-293 LRRK2 WT cells and KO cells were stained with Hoechst 33342 (**D**,**E**), primary antibodies against total-LRRK2 (**F**,**G**), and merged (**H**,**I**). Positive blue fluorescence reflects nuclei, and positive red fluorescence reflects total-LRRK2. (**J**) Quantification of the mean fluorescence intensity (MFI) in WT and KO cells. The figures represent one out of three independent experiments. One-way ANOVA, post hoc test Bonferroni. The data are expressed as the mean ± SD; *** *p* < 0.001. Image magnification, 200×.

**Figure 2 ijms-24-10474-f002:**
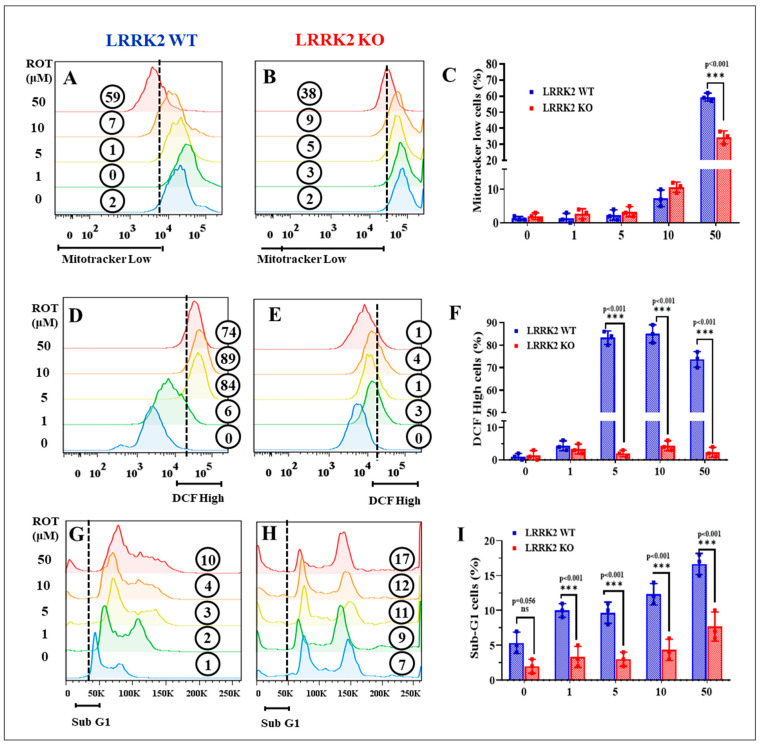
LRRK2 KO induces no significant ΔΨm damage, ROS production, or nuclear fragmentation. (**A**,**B**) Representative histograms showing the percentage of MitoTracker flow cytometry (FC) analysis from untreated HEK-293 LRRK2 WT and HEK-293 LRRK2 KO cells or cells treated with ROT (1, 5, 10, and 50 µM) at 6 h at 37 °C. (**C**) Percentage of MitoTracker in LRRK2 WT and LRRK2 KO untreated or treated with ROT. (**D**,**E**) Representative histograms showing the percentage of DCF flow cytometry analysis from untreated HEK-293 LRRK2 WT and LRRK2 KO cells or cells treated with ROT. (**F**) Percentage of DCF in LRRK2 WT and LRRK2 KO untreated or treated with ROT. (**G**,**H**) Representative histograms showing the percentage of SubG1 flow cytometry analysis from untreated HEK-293 LRRK2 WT and LRRK2 KO cells or cells treated with ROT. (**I**) Percentage of SubG1 in LRRK2 WT and LRRK2 KO untreated or treated with ROT. The image represents one of three independent experiments. The data are presented as the mean ± SD of three independent experiments (n = 3). Two-way ANOVA followed by Bonferroni’s test: Statistically significant differences when *** *p* < 0.001. ns: no significance. (**J**–**Y**) Representative fluorescence microscopy photographs showing untreated LRRK2 WT cells and KO cells or treated with ROT (50 μM) for 6 h and stained with MitoTracker (**J**–**M**), DCFH2DA (**N**–**Q**), Hoechst (**R**–**U**), and merged (**V**–**Y**). Positive red fluorescence reflects mitochondrial membrane potential (ΔΨm), positive green DCF fluorescence reflects the cytoplasmic presence of reactive oxygen species (ROS), and positive blue fluorescence reflects nuclei. (**Z**,**AA**) Quantification of the Mitotracker and DCFH2DA mean fluorescence intensity (MFI), respectively, in LRRK2 WT and KO cells. The figures represent one out of three independent experiments. One-way ANOVA, followed by Tukey’s test. Statistically significant differences when *** *p* < 0.001. Image magnification, 200×. The image represents one out of three independent experiments (n = 3).

**Figure 3 ijms-24-10474-f003:**
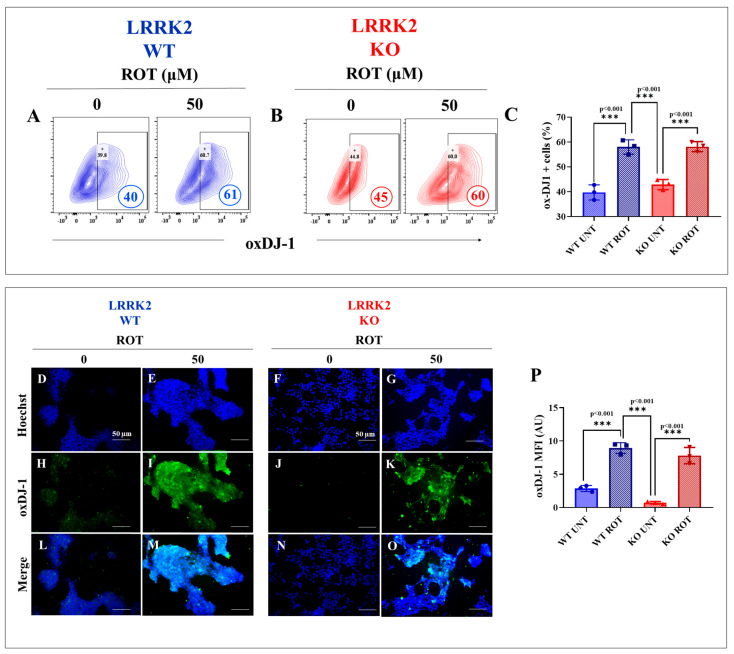
LRRK2 KO shows almost no oxidation of DJ-1 in the presence of ROT. (**A**,**B**) Representative flow cytometry contour plots showing oxidized protein DJ-1 (DJ-1 Cys^106^-SO_3_) in LRRK2 WT and LRRK2 KO untreated or treated with ROT (50 μM). (**C**) Percentage of oxDJ-1 (DJ-1 Cys^106^-SO_3_) in LRRK2 WT and LRRK2 KO cells untreated or treated with ROT. (**D**–**O**) Representative fluorescence microscopy photographs showing untreated LRRK2 WT cells and KO cells, or treated with ROT (50 μM) for 6 h and stained with Hoechst (**D**–**G**), primary antibodies against oxDJ-1 (**H**–**K**), and merged (**L**–**O**). Positive blue fluorescence reflects nuclei, and positive green fluorescence reflects the cytoplasmic presence of oxDJ-1 protein. (**P**) Quantification of the oxDJ-1 mean fluorescence intensity (MFI) in LRRK2 WT and KO cells. The figures represent one out of three independent experiments. One-way ANOVA, followed by Tukey’s test. The data are expressed as the mean ± SD; *** *p* < 0.001. Image magnification, 200×.

**Figure 4 ijms-24-10474-f004:**
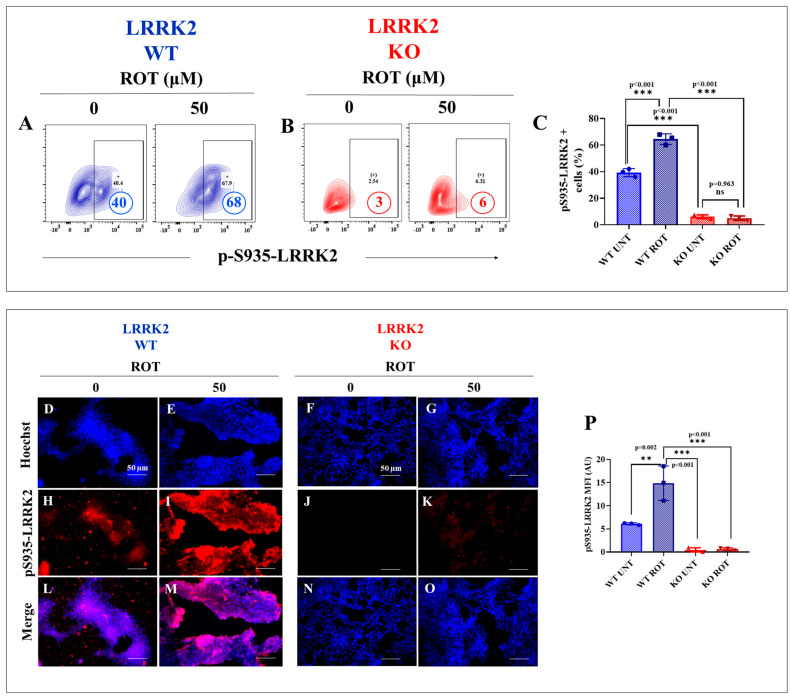
LRRK2 KO shows no significant p-S^935^-LRRK2 when exposed to ROT. (**A**,**B**) Representative flow cytometry contour plots show phosphorylated LRRK2 at residue S^935^ in LRRK2 WT and LRRK2 KO untreated or treated with ROT (50 μM). (**C**) Percentage of p-S^935^-LRRK2 in LRRK2 WT and LRRK2 KO untreated or treated with ROT. The data are presented as the mean ± SD of three independent experiments (n = 3). One-way ANOVA followed by Tukey’s test: statistically significant differences when *** *p* < 0.001. ns: no significance. (**D**–**O**) Representative fluorescence microscopy photographs showing untreated LRRK2 WT cells and KO cells or treated with ROT (50 μM) for 6 h and stained with Hoechst (**D**–**G**), primary antibodies against p-S^935^-LRRK2 (**H**–**K**), and merged (**L**–**O**). Positive blue fluorescence reflects nuclei, and positive green fluorescence reflects the cytoplasmic presence of p-S^935^-LRRK2 protein. (**P**) Quantification of the p-S^935^-LRRK2 mean fluorescence intensity (MFI) in LRRK2 WT and KO cells. The data are presented as the mean ± SD of three independent experiments (n = 3). One-way ANOVA followed by Tukey’s test: statistically significant differences when ** *p* < 0.01, *** *p* < 0.001. Image magnification, 200×.

**Figure 5 ijms-24-10474-f005:**
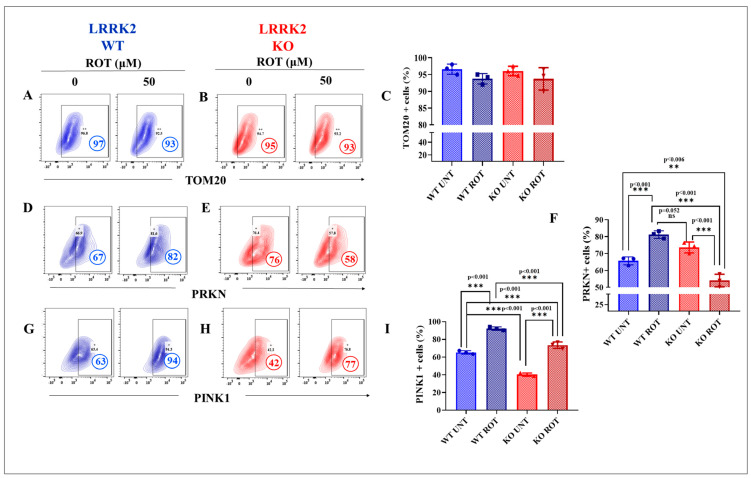
LRRK2 KO upregulates PINK1 but not PRKN under ROT stimuli. (**A**,**B**) Representative flow cytometry contour plots show HEK-293 LRRK2 WT and LRRK2 KO cells untreated or treated with ROT (50 µM) labeled with primary antibodies against TOM20. (**C**) Percentage of TOM20. (**D**,**E**) Representative flow cytometry contour plots show LRRK2 WT and LRRK2 KO cells untreated or treated with ROT (50 µM) labeled with primary antibodies against PRKN. (**F**) Percentage of PRKN. (**G**,**H**) Representative flow cytometry contour plots show LRRK2 WT and LRRK2 KO cells untreated or treated with ROT (50 µM) labeled with primary antibodies against PINK1. (**I**) Percentage of PINK1. The data are presented as the mean ± SD of three independent experiments (n = 3). One-way ANOVA followed by Tukey’s test: Statistically significant differences when ** *p* < 0.01, and *** *p* < 0.001; ns: no significance. In addition, HEK-293 WT LRRK2 cells and KO cells were stained Hoechst (**J**–**M**,**W**–**Z**), primary antibodies against PRKN (**N**–**Q**), and merged (**R**–**U**), PINK1 (**AA**–**AD**), and merged (**AE**–**AH**). Positive blue fluorescence reflects nuclei, positive green fluorescence reflects the cytoplasmic presence of PRKN protein, and positive red fluorescence reflects the presence of PINK1 protein. (**V**,**AI**) Quantification of the PRKN and PINK1 mean fluorescence intensity (MFI) in LRRK2 WT and KO cells. The figures represent one out of three independent experiments, followed by Tukey’s test: Statistically significant differences when * *p* < 0.05, ** *p* < 0.01, and *** *p* < 0.001. Image magnification, 200×.

**Figure 6 ijms-24-10474-f006:**
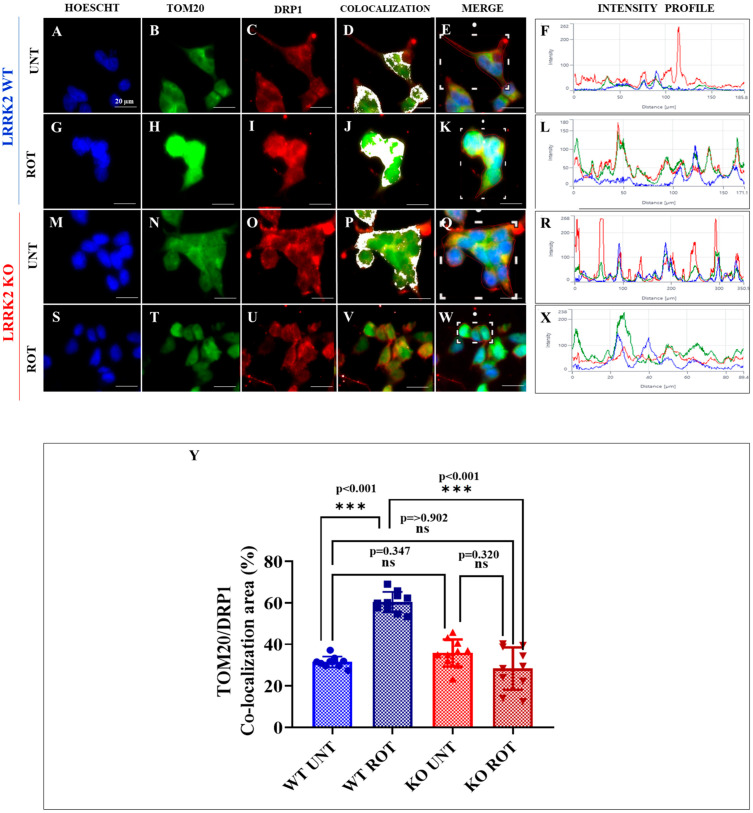
TOM20 and DRP1 do not colocalize in LRRK2 KO cells exposed to ROT. (**A**–**X**) Representative fluorescence microscopy photographs showing untreated LRRK2 WT cells and KO cells, or cells treated with ROT (50 μM) for 6 h that were labeled with Hoechst (**A**,**G**,**M**,**S**), primary antibodies against TOM20 (**B**,**H**,**N**,**T**), and DRP1 (**C**,**I**,**O**,**U**). Positive blue fluorescence reflects nuclei, positive green fluorescence reflects TOM20 protein, and positive red reflects TOM20 protein. Colocalization images (**D**,**J**,**P**,**V**) were used to select the merge image (broken square in panels **E**,**K**,**Q**,**W**) and calculate the intensity profile (**F**,**L**,**R**,**X**) by using Zen v.3.1 (Zeiss microscope software, Zeiss, Jena, Germany). Intensity (y-axis) was mapped against distance (μm x-axis) in the cell. The overlapping of green and red intensity histograms represents the colocalization of TOM20 and DRP1 (e.g., panel **L**). The percentage of positive colocalization (**Y**) was calculated as the colocalized area of each marker/total area on HEK293 LRRK2 WT and LRRK2 KO cells. The image represents one out of three independent experiments (n = 3). Experiments were performed in triplicate in three independent experiments. Dunnett’s T3 test. *** *p* < 0.001. ns: no significance. The data are presented as mean ± SD of three independent experiments. Image magnification, 400×.

**Figure 7 ijms-24-10474-f007:**
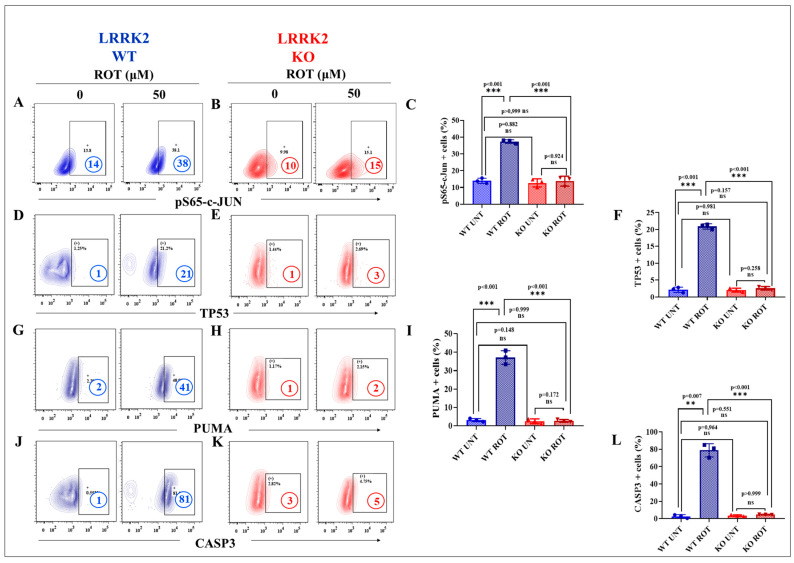
LRRK2 KO induces no activation of pro-apoptosis proteins under ROT stimuli. (**A**,**B**) Representative flow cytometry contour plots show HEK-293 LRRK2 WT and LRRK2 KO cells untreated or treated with ROT (50 µM) and stained with primary antibodies against c-JUN; (**C**) Percentage of pS^65^-c-JUN; (**D**,**E**) Representative flow cytometry contour plots show LRRK2 WT and LRRK2 KO cells untreated or treated with ROT (50 µM) and stained with primary antibodies against TP53. (**F**) Percentage of TP53. (**G**,**H**) Representative flow cytometry contour plots showing LRRK2 WT and LRRK2 KO cells untreated or treated with ROT (50 µM) stained with primary antibodies against PUMA. (**I**) Percentage of PUMA. (**J**,**K**) Representative flow cytometry contour plots show LRRK2 WT and LRRK2 KO cells untreated or treated with ROT (50 µM) stained with primary antibodies against CASP3. (**L**) Percentage of CASP3. The data are presented as the mean ± SD of three independent experiments (n = 3). One-way ANOVA followed by Tukey’s test or Dunnett’s T3 test: statistically significant differences when ** *p* < 0.01, and *** *p* < 0.001. ns: no significance. (**M**–**BK**) Representative fluorescence microscopy photographs showing untreated LRRK2 WT cells and KO cells or treated with ROT (50 µM) for 6 h and stained with Hoechst (**M**–**P**,**Z**–**AC**,**AM**–**AP**,**AZ**–**BC**), primary antibodies against p-S^65^-c-JUN (**Q**–**T**), merged (**U**–**Y**), TP53 (**AD**–**AG**), merged (**AH**–**AK**), PUMA (**AQ**–**AT**), merged (**AU**–**AX**), and CASP3 (**BD**–**BG**), and merged (**BH**–**BK**). Positive blue fluorescence reflects nuclei, positive red fluorescence reflects p-S^65^-c-JUN and TP53, and positive green fluorescence reflects PUMA and CASP3. Quantification of p-S^65^-cJUN (**Y**), TP53 (**AL**), PUMA (**AY**), and CASP3 (**BL**) mean fluorescence intensity (MFI) in WT and KO cells. The data are presented as the mean ± SD of three independent experiments (n = 3). One-way ANOVA followed by Tukey’s test: Statistically significant differences when ** *p* < 0.01, *** *p* < 0.001. Image magnification, 200×.

## Data Availability

All relevant data are within the manuscript and its Appendix A.

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
