# Peer review of "LRRK2 Knockout Confers Resistance in HEK-293 Cells to Rotenone-Induced Oxidative Stress, Mitochondrial Damage, and Apoptosis"

_ijms, 2023, doi:10.3390/ijms241310474_

Round 1

Reviewer 1 Report

Known in the field based on previous literatures:

1. Parkinson's disease (PD) is a progressive nervous system disorder that affects movement. PD is triggered by the selective loss of dopaminergic neurons in the substantia nigra (SN) region of brain.  The cause of PD is multifactorial among age, genetic and environmental factors are very important.

2. Both experimental and epidemiological data have reported pesticides and heavy metals exposure are the main environmental risk factors of PD.

3. Enzymatic and non- enzymatic pathways involved in free radical generation. Increased free radical production (reactive oxygen species and reactive nitrogen species), mitochondrial dysfunction and impaired antioxidant defense system are the key players contributing to oxidative stress in PD.

In this article authors reported following findings:

I have gone through the article titled "LRRK2 Knockout Confers Resistance of HEK-293 Cells to Rotenone-Induced Oxidative Stress, Mitochondrial Damage, and Apoptosis’. Authors created HEK-293 LRRK2 KO cells treated with rotenone, a well know pesticide and PD model, showed unaltered oxidative stress and apoptosis markers. Authors performed and reported following findings-

1. Authors created HEK-293 LRKK2 KO cells through CRISPR/Cas9 method and further validated its integration.

2. LRRK2 KO shows no oxidation of DJ-1 and significant phosphorylation in the presence of rotenone.

3. LRRK2 KO shows differential effect on PINK1 and PRKN. LRRK2 KO inhibits rotenone induced apoptosis.

The article presented are interesting and generally supportive of the conclusions drawn. There are, however, several issues including experimental validation which is not part of the article. The following minor suggestions if incorporated could help in the better understanding of the significance of the work and implications.

Minor Concerns:

1. How did you decide the dose of rotenone?  Have you measured IC50?

2. What is meaning of the sign in line 31, line 73 and at many places, rotenone (50—M)? Please write correctly everywhere.

3. In line 71, the word knockdown (KO) is confusing. Authors should write knock out (KO) not knock down.

4. Why there is base level difference of phosphorylation between WT UNT vs KO UNT?

5. Several studies are already available about association of LRKK2 in oxidative stress. Explain, how your article is different from rest and how does it address a specific gap in the field?

Reviewer 2 Report

The concentrations and units for some chemicals, treatments and results used in the manuscript are inapprehensible, resulting in difficulties to evaluate the data as well as the manuscript. 

Some data are inapprehensible.

Reviewer 3 Report

1) The manuscript needs minor correction, spelling, spacing etc.

Suggestion:

2) Western blot results of some proteins like caspase-3 etc can improve the quality of this research work.

3) Can include cell cycle analysis, apoptosis and necrosis cell death analysis.

Round 2

Reviewer 2 Report

Manuscript ID, ijms-2374808

Title: LRRK2 knockout confers resistance of HEK-293 cells to rotenone-induced oxidative stress, mitochondrial damage, and apoptosis.

This study used wild type and LRRK2 knockout HEK-293 cells to investigate effects of exposure to rotenone (ROT). LRRK2 KO cells treated with ROT showed unaltered OS and apoptosis markers. The study also found that LRRK2 acts as a proapoptotic kinase by regulating mitochondrial proteins, transcription factors, and CASP3 in cells under stress conditions. This study concluded that LRRK2 is an important kinase in the pathogenesis of PD. It is an interesting study.

Please find the following comments.

1). This study used CRISPR/Cas9 LRRK2 knockout (KO) in the human embryonic kidney 27 cell line 293 (HEK-293), and based on the results in this kedney cell line, the authors concluded LRRK2 is important in pathogenesis of PD. The characterization of this this cell line should be checked, for example, whether it expresses dopaminergic markers (TH, DAT) and synthesis dopaminergic transmitters (DA,L-DOPA ). The relationship between this kidney cell line and PD should be discussed, as well as the limitation of using this cell line as a model to interpret PD disease.

2). The realistic meaning of selected ROT concentration, i.e., 50uM, should be discussed, especially the connection with in vivo experiments of PD model.

3). To prove the role of LRRK2, this study used LRRK2 KO method. However, effects of ROT exposure on LRRK2 should also be checked, such as protein/mRNA expression levels, phosphorylation, or kinase activity. Furthermore, whether change of LRRK2 expression is observed or not in animal PD model and PD patients should also be discussed.

4). It is well understood that in the CNS, glial cells also play important roles in pathogenesis of neurodegenerative diseases, including PD, AD etc.  However, this study is based on a cell-autonomous hypothesis of apoptosis, i.e., the effects of ROT was checked only in HEK-293 cells, which was used as a neural like cell line. The design of this study might be very different from real in vivo condition, especially considering that glial cells play very important roles in oxidative stress, as well as pro-inflammatory responses. Limitations of this study should be discussed.

Minor editing of English language required.

Round 3

Reviewer 2 Report

The revised manuscipt is acceptable.